# Production of Human IFNγ Protein in *Nicotiana benthamiana* Plant through an Enhanced Expression System Based on *Bamboo mosaic* Virus

**DOI:** 10.3390/v11060509

**Published:** 2019-06-03

**Authors:** Min-Chao Jiang, Chung-Chi Hu, Na-Sheng Lin, Yau-Heiu Hsu

**Affiliations:** 1Ph.D Program in Microbial Genomic, National Chung Hsing University and Academia Sinica, Taichung 40227, Taiwan; hide410123@gmail.com; 2Graduate Institute of Biotechnology, Advanced Plant Biotechnology Center, National Chung Hsing University, Taichung 40227, Taiwan; cchu@dragon.nchu.edu.tw; 3Institute of Plant and Microbial Biology, Academia Sinica, Taipei 11529, Taiwan; nslin@sinica.edu.tw

**Keywords:** potexvirus, *bamboo mosaic virus*, interferon gamma, therapeutic protein, ER retention

## Abstract

Plant-based systems are safe alternatives to the current platforms for the production of biologically active therapeutic proteins. However, plant-based expression systems face certain major challenges, including the relatively low productivity and the generation of target proteins in biologically active forms. The use of plant virus-based expression systems has been shown to enhance yields, but further improvement is still required to lower the production cost. In this study, various strategies were employed to increase the yields of an important therapeutic protein, human interferon gamma (IFNγ), in *Nicotiana benthamiana* through modifications of expression vectors based on potexviruses. Among these, the vector based on a coat protein (CP)-deficient *Bamboo mosaic virus* (BaMV), pKB△C_His_, was shown to exhibit the highest expression level for the unmodified IFNγ. Truncation of the N-terminal signal peptide of IFN (designated mIFNγ) resulted in a nearly seven-fold increase in yield. Co-expression of a silencing suppressor protein by replacing the coding sequence of BaMV movement protein with that of P19 led to a 40% increase in mIFNγ accumulation. The fusion of endoplasmic reticulum (ER) retention signal with mIFNγ significantly enhanced the accumulation ratio of biologically active dimeric mIFNγ to 87% relative to the non-active monomeric form. The construct pKB19mIFNγER, employing the combination of all the above enhancement strategies, gave the highest level of protein accumulation, up to 119 ± 0.8 μg/g fresh weight, accounting for 2.5% of total soluble protein (TSP) content. These findings advocate the application of the modified BaMV-based vector as a platform for high-level expression of therapeutic protein in *N. benthamiana*.

## 1. Introduction

Plant molecular farming (PMF) is a cost-effective technology that uses plants to produce valuable pharmaceutical agents, such as therapeutic proteins, antibodies, enzymes, and edible vaccines [1,2,3]. Currently the plant-based systems are emerging as an alternative bio-production system to microbial and animal expression systems for the advantages in many aspects [3,4,5,6]. In particular, plant-based production systems offer the potential for low risk of animal pathogen contamination, eukaryotic post-translational modification, direct oral consumption of the expressed protein, low cost of biomass production, and rapid scalability [3]. Thus, plants may be an ideal expression platform to produce biologically active and safe pharmaceuticals at lower cost [7]. However, the low yields in some plant production systems have posed a significant challenge to the application of PMF-based pharmaceuticals at an industrial scale [8,9,10].

One of the solutions to the low-yield problem of PMF is by using plant virus-based systems in which the production efficiency can be improved (greater yields within a shorter time frame) through the effective infection and rapid replication of the viruses [11,12]. In addition, the use of plant virus-based vectors for transient expression may circumvent the concern over genetically modified organisms (GMO) [13]. Consequently, many plant virus-based vectors have been applied in the production of valuable pharmaceutical proteins, as extensively reviewed previously (e.g., [11,13,14,15,16,17].

In addition to the yield requirement, certain therapeutic proteins pose another challenge for PMF: biologically active form of the target protein (TP). An ideal production system should support the efficient production of therapeutic proteins in forms that are biologically active. For example, the biologically active form of human interferon gamma (IFNγ), which is a highly valuable therapeutic protein because of its clinical applications and involvement in the immune system [18,19], is a dimer (D) generated by anti-parallel interlocking of two monomers (Ms) of 17 kDa each [20]. Many researchers have explored heterologous expression platforms and used different methods to produce the active dimeric IFNγ, including chimerical cross-linking of the M IFNγ produced in *baculovirus-* or *E. coli-*based expression systems [21,22,23,24,25,26]. Thus, it is necessary to verify whether the desired protein expression system facilitates the production of dimeric IFNγ (D IFNγ). The proportion of D IFNγ relative to the total IFNγ yield is therefore the key determinant of the economic value of the production system.

*Bamboo mosaic virus* (BaMV), a member of the genus Potexvirus, has also been developed into various vector systems by our group for applications such as virus-induced gene-silencing (VIGS) [27], expression of epitopes on chimeric BaMV particles as vaccine candidates [28,29,30,31], and production of fluorescent antibody-labeling detector protein [32] in plants. However, the yields of these systems require further improvement to be economically competitive, and the ability to produce TPs in their biologically active forms, such as Ds, needs to be verified.

In this study, we aimed to improve the yields of the potexvirus-based vector system through various strategies, including the usage of different potexviruses as the backbone, truncation or fusion of various signal tags, and the co-expression of different RNA silencing suppressors. As a model system, IFNγ was selected as the TP in this study, with an additional attempt to evaluate the ability and efficiency of the BaMV-based system in producing IFNγ in the D form. The results revealed that the yield improvement and effective production of D IFNγ could be achieved by the combination of several approaches tested in this study.

## 2. Materials and Methods

### 2.1. Constructions

A previously constructed infectious clone of BaMV (GenBank accession no. AF018156.2) [27], pKB, containing the full-length cDNA of BaMV-S strain under the control of a dual constitutive 35S promoter of *Cauliflower mosaic virus* and a nopaline synthase (nos) terminator was used as the starting material for all constructs and also served as the positive control in the following analyses. The containment of the viral vectors is a critical concern in the production of recombinant proteins in plants. Since CP is required for cell-to-cell movement of potexviruses [33,34], we have constructed potexvirus-based viral vectors by replacing the CP open reading frame (ORF) with that of the TP in order to prevent the undesired spread of the viral vectors. It has been previously shown that nucleotides (nts) +1 to +15 at the 5′-terminal of BaMV-S strain CP ORF are essential for CP subgenomic RNA promoter activity (SGP) [35]. Thus, a pKB-derived vector, pKB△C_His_, was generated, containing multiple cloning sites (*Mlu*I-*Stu*I-*Not*I*-Spe*I) and 6X His tag to facilitate the purification of TP and nts +1 to 15 at the 5’ terminus of BaMV CP gene, of which initiation codon was mutated. Details for the construction of pKB△C_His_ are described in the Appendix A. The IFNγ DNA (GenBank accession no. AY121833.1) fragment was amplified with full-length IFNγ gene as template (from Lin’s Lab) using gene specific primer F-IFNγ-*Mlu*I plus R-*Spe*I-IFNγ (Appendix A) The amplified PCR fragment was digested with *Mlu*I and *Spe*I, and ligated into the pKB△C _His_ vector restricted with cognate enzymes to generate the recombinant plasmid pKBIFNγ_._


Previous reports have demonstrated that potexviruses, such as PVX and FoMV, are good candidates for the development of virus-mediated overexpression vectors for rapid production of heterologous recombinant [36,37,38,39]. As an initial attempt to improve the IFNγ production efficiency, we compared the yields of BaMV-, PVX-, or FoMV-based vector systems for the transient expression of IFNγ. For this purpose, the other CP deficient potexvirus-based vectors, pKPIFNγ and pKFIFNγ, were constructed using similar strategy, with the respective CP coding region replaced by that of with IFNγ using specific enzyme digestions, *Mlu*I and *Spe*I for *Potato virus X* (PVX) (GenBank accession no. AF272736.2) and *Mlu*I and *Not*I for *Foxtail mosaic virus* (FoMV) (GenBank accession no. AY121833.1).

Human IFNγ cDNA is a single polypeptide consisting of 166 amino acids, 20 of which at the N-terminus constitute a characteristic signal peptide [40]. The mature IFNγ (mIFNγ, without native signal peptide) polypeptide composed of 146 amino acids contains the major biologically active region [41]. Our previous attempt to over-express the full-length IFNγ in *E. coli* system did not result in appreciable amount of the TP. However, after truncating the coding region of the N-terminal signal peptide, the mIFNγ could be over-expressed successfully, which was consistent with the previous studies [23,24]. To test the feasibility of this approach in BaMV-based vector system in plants, the DNA fragment was amplified with full-length IFNγ gene as template (from Lin’s Lab) using gene specific primer pair F-*Mlu*I-*Nco*I-mIFNγ plus R-*Not*I -TGA-His-IFNγ (Appendix A). The amplified PCR fragment was digested with *Mlu*I and *Not*I, and ligated into the pKB△C _His_ vector restricted with the same enzymes to generate the recombinant plasmid pKBmIFNγ, in which the native signal peptide of IFNγ was truncated to give mIFNγ. 

It has been shown that IFNγ protein expression can be significantly enhanced in CHO cell-based system [42] and also in *Pichia pastoris* cell-based system [43] through optimizing the codons of IFNγ. To further increase translation efficiency by codon optimization based on *N. benthamiana* codon bias, the synthesis of IFNγ-1 or IFNγ-2 gene carrying the optimized codons (the full-length sequences of which were shown in Appendix A) was performed by Protech company (Protech, Taipei, Taiwan). The DNA fragments were amplified with IFNγ-1 or IFNγ-2 gene as templates using specific primer pairs F:*Mlu*I-ATG-IFNγ-1/C-2 plus R-*Not*I-TGA-His-IFNγ and F:*Mlu*I-*Nco*I-mIFNγC-1 plus R-*Not*I-TGA-His-IFNγ, respectively, as listed in Appendix A. The amplified PCR fragment was digested with *Mlu*I and *Not*I and ligated into the pKB△C _His_ vector restricted with the same enzymes to generate the recombinant plasmid pKBIFNγ-1, pKBIFNγ-2, pKBmIFNγ-1, and pKBmIFNγ-2. 

Previous reports showed that the co-expression of viral silencing suppressor proteins, P19 or P38, resulted in high-level accumulation of chimeric BaMV particles expressing VP1 epitope in transgenic *N. benthamiana* plants and suspension cell cultures [31]. To insert different gene silencing suppressors, P19, P28, or P38 [44,45] into BaMV cassettes, the constructs established previously [31] were used as the templates for modifications. By replacing the coding regions of TGBp1–3 of pKBmIFNγ with those for silencing suppressors P19 (519 bp, GenBank accession no. AJ288926), P28 (762 bp, GenBank accession no. AAB70563.2), or P38 (1056 bp, GenBank accession no. HQ589261), respectively, the constructs of pKB19mIFNγ, pKB28mIFNγ, and pKB38mIFNγ were generated. The coding sequence of P19, P28, and P38 were amplified with specific primer pairs containing *Dra*Ш restriction enzyme site as listed in Appendix A. The plasmid pKBmIFNγ was digested with *Dra*Ш to remove most of the TGBp1–3 region from this construct and different PCR products of P19, P28, or P38 were inserted at the same site. 

It has been reported that the fusion of arabinogalactan-protein (AGP) signal of ten Ser-Pro dipeptide repeats, (SP)_10_, or ER retention signal peptide Ser-Glu-Lys-Asp-Glu-Leu to the C-terminus of TP could stabilize its accumulation [46,47,48]. Thus, the strategy was tested on the BaMV-based vector system in this study. AGP signal was constructed by primer extension of two mutually complementary primer oligonucleotides, F-SP-*Spe*I and R-SP-*Spe*I (Appendix A), followed by digestion with *Spe*I. The digested product was cloned into pKBmIFNγ to generate the recombinant plasmid pKBmIFNγ(SP)_10_. The ER retention signal was amplified using a specific reverse primer, containing coding sequencing of Ser-Glu-Lys-Asp-Glu-Leu, R-*Spe*I-SEKDEL-IFNγ and a forward primer, F-*Mlu*I-*Nco*I-mIFNγ (Appendix A) with a plasmids pKBmIFNγ as the template, and digested with *Mlu*I and *Spe*I. The digested product was cloned into pKB△C_His_ to generate the recombinant plasmid pKBmIFNγER. To obtain the optimal BaMV-based vector, the plasmid pKBmIFNγER was used as a template for the replacement of TGBp1–3 region by the coding sequences of silencing suppressors P19 to generate pK19mIFNγER. All constructs were confirmed by sequencing and transformed into *Agrobacterium tumefaciens* strain pGV3850 via electroporation.

### 2.2. Transient Expression of TP in N. benthamiana Plants

Transient expression in *N. benthamiana* was performed by *Agrobacterium*-mediated infiltration. *A. tumefaciens* (PGV3850) clones harboring different constructs expressing TP were grown separately. Cells were harvested by centrifugation at 12,000 rpm and resuspended in *agroinfiltration* buffer (10 mM MgCl_2_ and 10 mM MES) to achieve an OD_600_ of 0.5 for each construct. Then the culture was infiltrated into 6-week-old *N. benthamiana* plants (at the 5–6 leaf stage) by the use of a 1 mL syringe without a needle. The infiltrated plants were maintained in the greenhouse at 28 °C and 16-h light/8-h dark.

### 2.3. Immunoblotting Assay

Total protein was extracted from agroinfiltrated leaves at 3 and 5 days-post infiltration (DPI) with use of 1:2.5 (*w/v*) protein extraction buffer (50 mM Tris-HCl, pH 8.0, 10 mM KCl, 10 mM MgCl_2_, 1 mM EDTA, 20% glycerol, 2% SDS, and 10% β-Mercaptoethanol). Extracted total proteins were separated by 12% SDS-PAGE for 1 h at 150 V. Following electrophoresis, the gels were either stained with Coomassie Brilliant Blue G-250 or transferred to Immobilon^TM^ PVDF membrane (Millipore, Billerica, MA, USA) for 1 h at 200 mA, which was then blocked with blocking solution (0.5% BSA, 5% nonfat milk, in 1× PBS. The blots were then incubated with 1:5000 dilution of rabbit primary antibodies against mIFNγ as shown in Appendix A, His tag, P19, P28, or P38, as described previously [31] at 37 °C for 1 h, followed by goat anti-rabbit IgG alkaline phosphatase secondary antibody (1:5000 dilution) at 37 °C for 1 h. The specific protein bands on the blots were finally visualized by NBT/BCIP color development (Thermo Scientific, Waltham, MA, USA) and the band intensities quantified by Image Reader LAS-4000. The expression level of IFNγ relative to that of RubisCO small subunit was determined from band intensities of coomassie blue staining (CBS) and immunoblot (IB) analysis. All measurements were performed in triplicates.

### 2.4. Northern Blot Analysis

Total RNA extracted from agroinfiltrated leaves at 3 and 5 DPI was analyzed following the classic procedure as described [49]. After denaturing with glyoxal, RNAs were separated by electrophoresis and transferred onto nylon membrane (Amersham, Buckinghamshire, UK). To detect RNA levels of BaMV and chimeric BaMV, the blots were hybridized with ^32^P-labeled probes specific to (+)-strand BaMV RNA as described previously [50].

### 2.5. Quantitative ELISA

For quantification of TP expression levels by different chimeric BaMV vectors, agroinfiltrated leaves at 3 and 5 DPI were analyzed by enzyme-linked immunosorbent assay (ELISA) as described previously with minor modifications [31]. Total protein samples were prepared from inoculated leaves with the use of 1:5 (*w/v*) coating buffer (0.1 M carbonate/bicarbonate buffer, pH 9.6). After standing for 20 min, the supernatant was recovered and quantified for total soluble protein using Bradford colorimetric assay (Sigma-Aldrich, St. Louis, MO, USA). The concentration of TSP in every sample was adjusted to approximately 4.5–5 mg/mL. The 96-well microtiter plates were coated with different dilutions of protein extracts from non-inoculated and inoculated leaves or purified mIFNγ protein derived from *E. coli* for standard curve and the positive control, and then incubated at 37 °C for 1 h. After washing three times with PBST (0.05% Tween 20, in 1× PBS), 100 μL blocking buffer (5% skim milk, in 1× PBS) was added to each well. The plate was incubated at 37 °C for 1 h, washed and incubated with rabbit polyclonal anti-mIFNγ antibody (100 μL/well, diluted 1:500 in blocking buffer) at 37 °C for 1 h. After washing, alkaline phosphatase-conjugated goat anti-rabbit IgG (Jackson Immuno Research, 100 μL/well, diluted 1: 2000 in blocking buffer) was added, and the plate was incubated at 37 °C for 1 h. After washing, TP quantification was measured after 2 h at 405/490 nm on microplate reader (Spectramax M2; Molecular Devices, Sunnyvale, CA, USA) using p-nitrophenyl phosphate solution (Sigma, USA, 100 μL/well, 1 tablet in 10 mL Diethanolamine buffer, pH 9.8). The concentration of TP was determined by comparison with known amounts of the purified mIFNγ protein derived from *E. coli*. All measurements were performed in triplicates.

### 2.6. Statistical Analysis

Data obtained from three replicate samples are expressed as mean ± SD. Statistical analysis was performed using ANOVA. *p*-value of <0.001 was considered significant.

## 3. Results

### 3.1. Construction of Chimeric BaMV Expression Cassettes

For biocontainment of the recombinant constructs, we generated a chimeric BaMV viral vector, pKB△C_His_, by deleting most of the CP coding sequence while retaining the CP SGP region, with the addition of 6× His tag for purification of TP. This cassette was placed under the control of dual constitutive 35S promoter of *Cauliflower mosaic virus* (CaMV) and a nopaline synthase (nos) terminator as shown in Appendix A). To develop a BaMV-based vector for human IFNγ protein expression, a full-length cDNA sequence of IFNγ was cloned into pKB△C_His,_ to generate pKBIFNγ (Figure 1A). When pKBIFNγ was infiltrated into *N. benthamiana* leaves, no visible symptoms were observed. At 3, 5, and 7 DPI, samples were collected from the infiltrated *N. benthamiana* leaves and assayed for protein expression by western blot with specific antibody against mIFNγ (antigen derived from *E. coli*). The result indicated that IFNγ could be produced successfully (Figure 1B and Figure 2B). 

### 3.2. Effect of Different CP-Deficient Potexvirus-Based Vectors on Yields of IFNγ

To compare the performance of different potexvirus-based vectors, plasmids pKPIFNγ and pKFIFNγ were constructed based on CP-deficient PVX and FoMV, respectively, using similar strategy as that used for pKBIFNγ construction (Figure 1A), and infiltrated into *N. benthamiana* leaves through *Agrobacterium*-mediated inoculation. The infiltrated leaves were collected at 3, 5, and 7 DPI and assayed for IFNγ protein yield by western blot and ELISA using specific IFNγ antibodies. The result of western blot analysis showed that IFNγ can be produced by these three potexvirus-based vectors (Figure 1B) and the accumulation level reached plateau at 5 DPI (Figure 1B,C). Among the three constructs, BaMV-based vector supported the highest expression level of IFNγ, reaching 8.3 ± 1.26 μg/g fresh weight, and did not cause severe degradation of IFNγ at 7 DPI, as compared to other viral vectors (Figure 1B,C). Therefore, the BaMV-based vector system was selected for further modifications and improvements in the following experiments.

### 3.3. Effect of N-Terminal Truncation and Codon Optimization of IFNγ

To test the effect of N-terminal signal peptide and codon optimization on the yield of TP, we truncated the signal peptide of IFNγ, generating a construct pKBmIFNγ, and further modified it by codon optimization based on the *N. benthamiana* codon bias. The codon-optimized cDNAs, IFNγ-1/IFNγ-2, and mIFNγ-1/mIFNγ-2, were synthesized and cloned into pKBIFNγ and pKBmIFNγ expression cassettes, respectively. Western blot analysis of total proteins from *N. benthamiana* leaves infiltrated with *A. tumefaciens* harboring pKBIFNγ and pKBIFNγ-1 revealed the presence of two distinct bands with apparent molecular weights (*Mr*) of 20.3 and 22.3 kDa due to 1, 2 N-glycan present in the mIFNγ, and an additional band with apparent *Mr* between 36.6 and 44.6 kDa, probably due to dimerization of mIFNγ proteins (Figure 2A,B). On the other hand, those infiltrated with *A. tumefaciens* harboring pKBIFNγ-2 exhibited an extra protein of 18.3 kDa and had a higher protein accumulation level compared with those infiltrated with pKBIFNγ and pKBIFNγ-1 (Figure 2A,B). In contrast to the banding patterns in the western blot of pKBIFNγ, pKBIFNγ-1, pKBIFNγ-2, the mature, non-glycosylated forms pKBmIFNγ, pKBmIFNγ-1 and pKBmIFNγ-2 exhibited two distinct bands with estimated *Mr* of 18.3 and 36.6 kDa, respectively, which were more apparent (Figure 2B). Previous studies reported that the protein species with *Mr* of 18.3, 20.3, and 22.3 kDa represent monomeric IFNγ/mIFNγ (M IFNγ/mIFNγ) occupied with 0, 1, or 2 N-glycan, respectively; whereas the protein with *Mr* of 36.6 kDa and those with *Mr* of 36.6 to 44.6 kDa and 36.6 kDa represent homo- and hetero-dimers of IFNγ/mIFNγ (D IFNγ/mIFNγ), respectively [22,24,51]. 

To clarify the expression level of IFNγ in different samples, we quantified the intensity of each band of IFNγ species relative to that of small subunit RubisCO. Densitometric quantification result revealed that leaves infiltrated with pKBmIFNγ (5 DPI) exhibited at least 7 times increase in IFNγ expression level compared to those infiltrated with pKBIFNγ-2 (Figure 2B,C). However, the IFNγ protein accumulation level in leaves infiltrated with pKBmIFNγ-1 and pKBmIFNγ-2 were slightly lower than that infiltrated with pKBmIFNγ (Figure 2B,C). For comparison of BaMV RNA accumulation level in different treatments, pKBmIFNγ infiltrated leaves also exhibited higher viral RNA accumulation at 5 DPI than those in leaves infiltrated with other BaMV expression cassettes (Figure 2D). 

The only difference between IFNγ and mIFNγ is the N-terminal signal peptide, consisting of 20 amino acids. To verify that the higher signal intensity for mIFNγ was not resulted from the preferences of the primary antibody, we have performed immunoblot analyses using different primary antibodies with different specificity, including anti-IFNγ (ab133566 from abcam Co., specific to N-terminal 1–100 amino acids of native IFNγ), anti-His tag (GTX115045 from GeneTex Co., specific to the 6X His tag), and the original anti-mIFNγ (Appendix A). The result revealed that all three primary antibodies showed no preference for IFNγ and mIFNγ. In addition, based on the band intensities of mIFNγ 2-fold serial dilutions, mIFNγ protein accumulation is about 8-fold higher than that of IFNγ (Appendix A).

In addition, the result of transient expression assay showed that IFNγ protein accumulation level in leaves infiltrated with pKBmIFNγ was 12 times higher than that in those infiltrated with pKmIFNγ (nonviral vector system) at 5 DPI (Figure 3B,C). The mIFNγ protein at 18.3 kDa was clearly observed even with the less sensitive CBS (Figure 3B upper panel). The result indicated that BaMV-based expression construct pKBmIFNγ dramatically improved TP accumulation in infiltrated *N. benthamina* leaves, further supporting the usage of plant-viral vector to produce recombinant proteins in plants. The result also showed that the constructed pKBmIFNγ led to the highest-level protein accumulation in infiltrated *N. benthamiana* leaves and was thus used for further optimization in the following experiments. 

### 3.4. Effect of Viral Silencing Suppressors

To test whether different viral silencing suppressors could improve TP accumulation by using CP-deficient BaMV-based vectors, we further modified the construct pKBmIFNγ by replacing the ORFs of the triple-gene-block proteins with those of different viral silencing suppressors, P19, P28, or P38 [44,45], so that the IFNγ and viral silencing suppressor could be co-expressed from the same construct. Northern blot analysis revealed that viral genomic RNA (gRNA) and subgenomic RNA (sgRNA) accumulation in leaves infiltrated with pKB19mIFNγ and pKB38mIFNγ were significantly increased compared to those with pKBmIFNγ, pKB28mIFNγ, and pKB△C_His_ (as a negative control) at 3 and 5 DPI (Figure 4D). The results of western blot analysis and densitometric quantification showed that the mIFNγ accumulation in leaves infiltrated with pKB19mIFNγ was significantly improved, up to approximately 40% compared to other constructs (Figure 4B,C). To further confirm the expression of P19, P28, and P38, western blot analysis was performed with specific antibodies against each viral silencing suppressor. The result verified that all viral suppressors could be stably expressed at either 3 or 5 DPI (Figure 4B). The above result indicated that P19 may serve as the suitable silencing suppressor for the CP-deficient BaMV-based vector to further improve the yields of target proteins.

### 3.5. Effect of Fusing ER Retention

It was found that IFNγ or mIFNγ produced by BaMV-based expression cassettes in *N. benthamiana* could fold naturally into Ds (Figure 1B, Figure 2B, and Figure 3B). In order to further improve the yields of D mIFNγ, we generated constructs pKB19mIFNγER and pKB19mIFNγ(SP)_10_ by fusing ER retention or AGP signal (SP)_10_ at the C-terminus of IFNγ to increase the stability of the recombinant protein [46]. Western blot analysis with specific antibody against mIFNγ showed that leaves infiltrated with pKB19mIFNγ, pKB19mIFNγER, and pKB19mIFNγ(SP)_10_ could produce both M and D form of IFNγ at 5 DPI (Figure 5B). We also clearly observed two distinct bands with *Mr* of 18.3 and 20.2 kDa, representing mIFNγ, and mIFNγER, respectively, by CBS (Figure 5B, in upper panel). The amounts of M and D form were further quantified by densitometry, as shown in Figure 5B,C. We found that the fusion of ER retention signal (pKB19mIFNγER) could enhance significantly the amount of D IFNγ compared with those infiltrated with pKB19mIFNγ and pKB19mIFNγ(SP)_10_ (Figure 5C), but the RNA accumulation of chimeric BaMV was not affected (Figure 5D). In addition, the D/M ratio of IFNγ produced in leaves infiltrated with pKBmIFNγER at 5 DPI was increased up to 87 %, which is higher than that from leaves infiltrated with pKB19mIFNγ and pKB19mIFNγ(SP)_10_ (51%), and that produced by *E. coli* (10%) mIFNγ without chemical cross-linking (Figure 5B, bottom panel). 

The overall accumulation of various species of IFNγ were further quantified using ELISA. The result revealed that the average levels of IFNγ produced in leaves infiltrated with pKB19mIFNγER and pKB19mIFNγ was increased to 119 ± 0.8 and 111± 5.8 μg/g fresh weight, accounting for 2.5% and 2.3% of TSP, respectively (Table 1). The results demonstrated that by fusion of either ER retention or AGP signal peptides, the IFNγ protein accumulation level was increased up to approximately 29% and 20%, respectively, as compared to that produced in leaves infiltrated with pKB19mIFNγ. These results demonstrated that, by targeting the recombinant protein mIFNγER into ER, higher ratio of D IFNγ could be produced. Therefore, through a combination of different approaches, including selection of suitable potexvirus vector backbone, truncation of signal peptide, codon optimization, co-expression of viral silencing suppressors, and the fusion of ER retention signal, it was shown that the optimal yield of biologically active dimeric IFNγ proteins could be achieved. The construct pKB19mIFNγER may serve as a practical vector for the production of IFNγ in plants with the highest yield.

## 4. Discussion

The expression efficiency of heterologous protein production systems is affected by a variety of factors [52,53]. In this study, we contemplated specific factors pertaining to the viral expression systems for plant production platforms, including the effects of different CP-deficient potexvirus-based vectors, TP sub-region for expression, TP codon optimization, co-expression of silencing suppressors, fusion of ER retention signal, and oligomeric TP stability. Accordingly, a series of BaMV-based IFNγ expression constructs were generated and assayed for their expression efficiency. The results revealed that the construct pKB19mIFNγER provided the highest yield of D IFNγ in infiltrated *N. benthamiana* leaves. 

For better biocontainment, all constructs used in this study were based on the CP-deficient viral vectors, since CP is required for the movement of potexviruses [54,55,56]. The lack of CP may have reduced the stability and accumulation levels of such viral vectors. However, such deficiency could be compensated by constructing transgenic plants harboring such viral vectors, in which every cell could express the viral vector that could then replicate autonomously to increase the yield of TP. Moreover, the transgenic plants could be further used to develop suspension cell culture systems for the production of TP complying with the Current Good Manufacture Practice (cGMP) regulations, as shown by our previous study [31].

In this study, we initially compared the IFNγ accumulation levels using different CP-deficient potexvirus-based vector systems (BaMV, PVX, and FoMV), and found that the BaMV-based vector provides the highest IFNγ expression level (Figure 1B,C). The difference in the yields of IFNγ by various potexvirus-based vector could be attributed to the differences in the replication/translation efficiency of these viruses in *N. benthamiana* or the difference in the stability of the same target protein (IFNγ) in plants infected by different potexviruses, since different viruses may elicit different defense responses in the same host plant. It has been reported that the expression level of a target protein, Flg-4M, through a PVX-based system could account for as high as 30% of TSP (~1 mg/g fresh leaf tissue) at 4 to 5 DPI, whereas the yield of the same target protein was only about one third of that when using a *Cowpea mosaic virus*-based vector [39]. These observations indicated that specific target proteins may require different viral vector systems to achieve optimal yields, and also reinforced the requirement to develop expression systems based on diverse viruses.

The nucleotides from +1 to +15 at 5′-terminal of CP gene were retained in the construct, since the region is the necessary enhancer elements for the CP SGP activity of BaMV-S strain. In contrast, another strain of BaMV, BaMV-O, requires extended nucleotides, from +1 to +52 for the SGP enhancer function [35]. The retention of enhancer elements for SGP function in the vector construct is crucial for potexvirus-based vectors. For example, the vector derived from *Pepino mosaic virus,* (PepMV) also retained some modulatory elements of CP gene to enhance SGP activity [57]. The PVX-based viral vector with deletion of 60 nts at 3’-terminus of CP gene has suffered from the drastic reduction of viral genome accumulation [58]. Based on the observation for PVX-based vector, we have also constructed another BaMV-based vector in which 3’-terminal nucleotides (positions 40–250) of CP coding region were retained between TP and BaMV 3’-UTR for the enhancement of TP expression. However, in the BaMV case, northern blot and western blot results showed that the retention of these 3’-terminal regions did not enhance BaMV replication and TP accumulation (Appendix A). The result indicated that different viruses, although belonging to the same genus, may have different requirement for expression enhancement of heterologous TPs.

As for the optimization of TP expression by selection of sub-region and codon usage, we truncated native signal peptide of IFNγ to give mIFNγ, and optimized the codon usage of IFNγ or mIFNγ according to that for *N. benthamiana* host. In these series of constructs, pKBmIFNγ could lead to the production of a stable mIFNγ at 3 and 5 DPI in infiltrated *N. benthamiana* leaves (Figure 2B). Based on the result shown in Figure 2B, the major differences that resulted in higher accumulation level of IFNγ-2 might be those at the N-terminal native signal peptide (Appendix A), since the truncation of the signal peptide led to similar accumulation levels of mIFNγ, mIFNγ-1, and mIFNγ-2, whereas in the presence of the signal peptide, IFNγ-2 accumulated to a much higher level compared to those of IFNγ and IFNγ-1. Such codon differences may affect protein translation efficiency resulting in higher accumulation of IFNγ-2 than IFNγ and IFNγ-1. For further improvement of mIFNγ accumulation, we included the coding sequence of viral silencing suppressor on the same vector for the expression of mIFNγ, and successfully increased the accumulation of chimeric BaMV RNAs (Figure 4C), which was in agreement with our previous result for chimeric BaMV particle expressing VP1 epitope [31]. In this study, the mIFNγ was produced as a separate protein, not fused to BaMV-CP, and the yield of mIFNγ was significantly improved up to 40% at 5 DPI (Figure 4B,C). This increase was comparable with that observed by using a PVX-based vector co-expressing GUS gene with silencing suppressor P19, which could increase expression up to 44% [59].

It was advantageous that IFNγ or mIFNγ expressed by the chimeric BaMV-based vector include not only the M but also the D form without chemical cross-linking (Figure 1B, Figure 2B, Figure 3B, Figure 4B, and Figure 5B). We could directly observe the D form TP under reducing and denaturation conditions of SDS-PAGE because IFNγ, and mIFNγ, is SDS-resistant protein and its polypeptide lacks intramolecular disulfide bonds [21]. In earlier studies, natural human IFNγ purified from fresh human peripheral blood lymphocytes existed simultaneously as M and D IFNγ [51]. There was also evidence that D IFNγ could bind to a specific cell surface receptor for exerting its biologically active function [60,61]. However, the vast majority of IFNγ and/or mIFNγ expressed by other expression systems, including *Oryza sativa* cells [62], *Brassica napus* seeds [46], *E. coli* [23,24], and *Pichia pastoris* [43], were largely Ms with hardly any Ds. D form of IFNγ or mIFNγ could only be generated through chemical cross-linking of the Ms produced in *baculovirus*- and *E. coli*-based systems [21,22,23,24]. In addition, the stability of the biologically active D IFNγ or mIFNγ is another major concern. Thus, we generated the construct pKBmIFNγER to alter mIFNγ subcellular localization into the ER compartment. We found that infiltration of pKBmIFNγER resulted in a higher proportion of D mIFNγ as compared to those produced by pKBmIFNγ or pKBmIFNγ(SP)_10_ (Figure 5B,C). The ratio of D/M reached to 87%, which is higher than the 10% observed in *E. coli* derived IFNγ without chemical cross-linking (Figure 5B, in the bottom panel). It has been reported that plant-derived rBChE with the fusion of ER retention signal could produce >95% of enzymatically active tetrameric form in *N. benthamiana* [63]. Another plant-derived immunoglobulin associated with ER-resident chaperon could correctly also form heterodimer in tobacco plants [64]. The ability of plant-derived recombinant proteins to correctly fold into mammalian-like active forms in the milieu of the ER could be attributed to three factors: (1) Plants have similar secretory mechanisms with mammals allowing for the transport of protein into ER compartment and the subsequent folding of protein into correct conformation [65,66]. (2) The ER possess certain chaperon proteins, such as BiP or calreticulin, which can assist in the assembly of protein oligomer [64]. (3) The ER provides a better environment with proper pH and fewer proteases, preventing protein degradation [46]. Therefore, we have taken this advantage to fuse the ER retention signal with mIFNγ, effectively increasing both the yield and the ratio of the biologically active D mIFNγ expressed by the modified BaMV-based vector (Figure 5) in infected *N. benthamiana*.

Plant virus-based vectors have been divided into two variants [67]: those with complete virus genomes capable of systemic movement and assembly of chimeric virus particles (full virus strategy) and the others with incomplete virus genomes (deconstructed virus strategy), such as the CP-deficient vectors which lacked virus CP and were prohibited from cell-to-cell or systemic movement. In addition to being environmentally friendly by prohibiting the cell-to-cell movement of the viral expression cassettes, the CP-deficient vectors can overcome insert size limitations and allow for the production of recombinant protein or antibody in large scale [36]. The representative of such system based on *Tobacco mosaic virus* (TMV), magnICON^®^, developed by Icon Genetics, Halle, Germany, has reported the yield of GFP reaching 5 mg/g fresh weight tissue in plants. Previous studies also have shown that co-expressing of viral RNA silencing suppressors, for example, *Tomato bushy stunt virus* P19 or *potato virus A* HcPro [58] in different plant viral expression systems can improve the production of recombinant proteins. In order to overcome the limitation on insertion size, magnICON^®^ vector was used in combination with a non-competing CP-deficient vector based on PVX to efficiently produce heavy and light chain of IgG, respectively. The individual chains of IgG were fused with ER-targeting signal to allow for proper assembly of heterotetramer mAbs, with yield reaching 0.5 mg/g tissue fresh weight in *N. benthamiana* plant [38]. In another case, ER-retained recombinant butyrylcholinesterase (prBChE-ER) can assemble into tetrameric form in plants by TMV RNA-based overexpression system [63]. Therefore, by the incorporation of different approaches, plant virus-based expression systems may serve as potential alternatives to express recombinant therapeutic proteins in plants.

With the combination of different approaches discussed above, the construct pKB19mIFNγER was shown to be the optimal BaMV-based expression vector for the efficient production of IFNγ or mIFNγ in *N. bethamiana*. The results of quantification by densitometry and ELISA showed that the infiltration of pKB19mIFNγER could lead to the production of the highest amount of TP, up to 119 ± 0.8 μg/g fresh weight, which corresponded to approximately 2.5% of TSP (Figure 5B,C, and Table 1). The yield is higher than those observed in previous studies for the production of IFNγ in plants with non-chloroplast expression system, including (1) rice suspension cells system, which produced human mIFNγ of 17.4 ng/mL media and 131.6 ng/g cell in culture medium and intracellularly, respectively [62]; (2) tobacco leaves system expressing chicken IFNγ of up to 10 to 20 μg/g fresh leaf weight [68]; and (3) a chimeric ZYMV-based vector system in *Chenopodium quinoa* leaves, which produced human mIFNγ of up to approximately 1–1.2 mg/100 g tissues [69]. It has been reported that the yield of a biologically active His-tagged GUS-IFNγ fusion protein could reach approximately 6% of TSP by using a transgenic tobacco chloroplast system [70]. However, the IFNγ produced in the chloroplasts existed in monomeric form, and the lack of proper glycosylation system in chloroplasts rendered the target protein non-glycosylated, which could be less stable and usually with lower biological activity [71,72]. In addition, the host range of BaMV covers many monocotyledonous plants, including wheat (*Triticum aestivum*), barley (*Hordeum vulgare*), rice (*O. sativa*), and *Brachypodium distachyon*. Previous studies showed that orally administered IFN still exhibits biological activity in humans and other animals [68,73,74,75]. It would be possible that human IFNγ expressed in edible plants through the BaMV-based vector developed in this study may be exploited orally to treat patients and animal without further purification from the plants. Thus, in comparison with similar systems previously reported for the expression of IFNγ, the BaMV-based vector has the following advantages: (1) better biocontainment of the vector, (2) higher yield relative to the non-chloroplast systems, (3) the capability of producing phosphorylated IFNγ in dimeric form, which were not supported by the chloroplast system, and (4) a broader host range in monocotyledonous plants. In contrast, the disadvantages of the BaMV-based vector system include the lack of movement ability of the CP-deficient BaMV vector and limited dicotyledonous hosts.

As a proof-of-concept study, the current research emphasized on developing an efficient BaMV-based vector for production of TP. The biological functions of the TP will be further corroborated in our upcoming studies. In addition, other factors affecting the biological activities of TP or the production efficiency will also be analyzed, such as the full glycosylation profiles of TP produced in plants and easy purification method for clinical investigation and industrialized production. For example, by using the fucosyl-transferase and xylosyl-transferase double knock-down lines (△FT/XT) of *N. benthamiana* [76,77], it is possible to reduce the immunogenicity and allergenicity of human TP produced in plants. In addition, we may alter the subcellular localization of TP by adding plant-derived signal peptides, leading target through an intact secretory pathway for decorated glycosylation [9,76,78]. An additional advantage of this approach is that the TP may be secreted into the extracellular space (apoplast space), from which could be obtained in higher amounts without complex purification. Through this TP secretion approach, it is also feasible to establish a plant suspension cell line to achieve a large-scale and continuous production a therapeutic protein.

## 5. Conclusions

In conclusion, we have explored the capability of a CP-deficient BaMV as an expression vector of a therapeutic protein, IFNγ, in *N. benthamiana* to increase the yield of IFNγ and to test the ability of such system to produce the biologically active IFNγ Ds. Through the combination of different optimization approaches, the construct, pKB19mIFNγER was generated, which led to the highest accumulation level of mIFNγ protein as the biologically active D form in the infiltrated *N. benthamiana* leaves. Our study demonstrated that the optimal yields of biologically active TPs could be achieved using viral vectors in plant-based molecular farming systems by the combination of different approaches, including (1) choosing the most suitable viral vector, (2) truncating unnecessary regions of the TP, (3) optimizing codon usages according to host system, (4) co-expressing viral silencing suppressor, and (5) relocating TP to proper subcellular compartments with fusion of signal peptides. Thus, our results advocate the use of CP-deficient BaMV-based vectors as an alternative system to efficiently produce biologically active therapeutic proteins in plants.

## Figures and Tables

**Figure 1 viruses-11-00509-f001:**
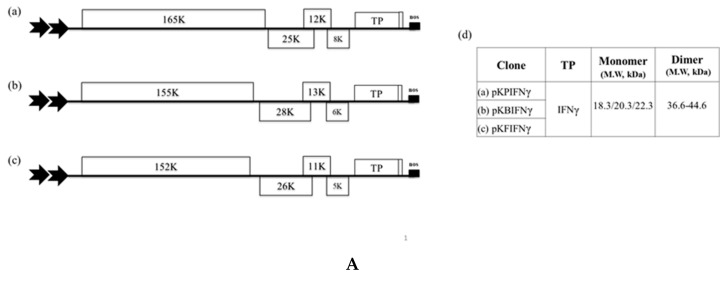
The expression of human interferon gamma (IFNγ) by different coat protein (CP)-deficient Potexvirus-based vectors in *N. benthamiana. (***A**). Schematic representation of Potexvirus-based expression cassettes, in which the CP coding regions of *Potato virus X* (PVX), *Bamboo mosaic virus* (BaMV), and *Foxtail mosaic virus* (FoMV) were replaced with that of the target protein (TP), IFNγ. These constructs were named pKPIFNγ (a), pKBIFNγ (b), and pKFIFNγ (c), with the expected molecular weight of the respective TP indicated in the adjacent table (d). (**B**). Analysis of TP expression in inoculated leaves. Total protein extracts, corresponding to 1 mg fresh weight in 1 mL of extraction buffer, were prepared from infiltrated leaf tissue at 3, 5, and 7 days-post infiltration (DPI) and analyzed by SDS-PAGE, followed by staining with CBS. IB analysis of proteins transferred to the membrane used anti-IFNγ as the primary antibody and goat-anti-rabbit IgG horseradish peroxidase conjugate as the secondary antibody. M, Marker; H, Healthy leaf sample; P, Positive control, purified mIFNγ protein derived from *E. coli* (100 ng in CBS and 2.5 ng in IB). The positions of various TP species and reference proteins were indicated on the right of the panel. The mono- and dimeric forms of IFNγ were indicated by M or D, respectively. (**C**). Quantification by ELISA. The ELISA quantification shows the IFNγ yield relative to fresh weight of leaves (μg/g). Statistical analysis was performed using ANOVA. *p*-value of <0.001 was considered significant. The double 35S promoters were indicated by the two thick arrows. The nos terminator is represented by the black box at the end of the construct.

**Figure 2 viruses-11-00509-f002:**
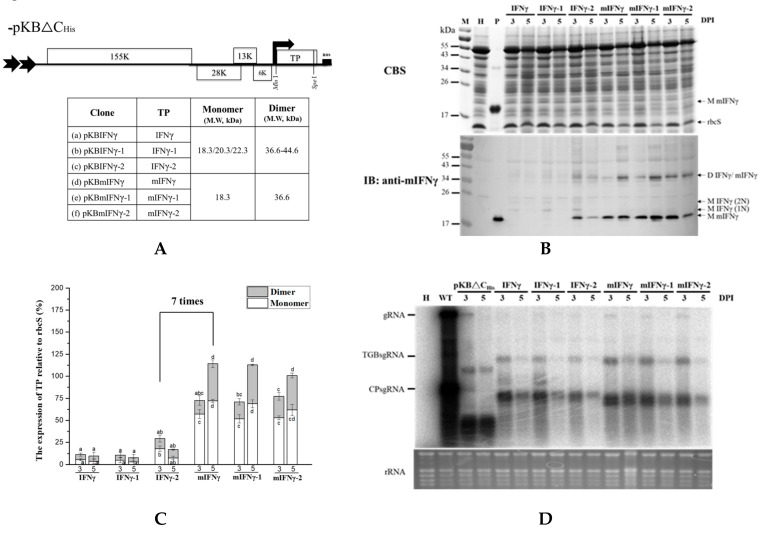
Codon usage optimization of IFNγ or mIFNγ gene based on the codon bias of *N. benthamiana*. (**A**). Schematic representation of *Bamboo mosaic virus* (BaMV)-based expression cassettes, in which BaMV CP coding region was replaced with TP (IFNγ or mIFNγ, for full-length or mature protein without signal peptide, respectively) using restriction sites *Mlu*I and *Spe*I. The coding sequence of either IFNγ or mIFNγ were further optimized according to the codon bias for *N. benthamiana* species to generate IFNγ-1/-2 and mIFNγ-1/-2, respectively. (**B**). Analysis of TP expression in inoculated leaves. Total protein extracts were prepared from infiltrated leaf tissue at 3 and 5 DPI and analyzed by SDS-PAGE, followed by staining with CBS and IB analysis with anti-mIFNγ as primary antibody and goat-anti-rabbit IgG alkaline phosphatase conjugate as secondary antibody. M, Marker; H, Healthy leaf; P, Positive control, purified mIFNγ protein derived from *E. coli* (250 ng for CBS and 25 ng for IB). (**C**). Quantification with densitometer. Densitometric quantification showing the percentage values of M or D TP relative to RuBisCO Small subunit, rbcS. Statistical analysis was performed using ANOVA. *p*-value of <0.001 was considered significant. (**D**). Northern blot analysis of wild-type or chimeric BaMV RNA in infiltrated leaves at 3 and 5 DPI. BaMV genomic RNA, and the subgenomic RNAs for triple gene block proteins (TGPsgRNA) and CP (CPsgRNA) were detected with a BaMV-specific probe, respectively (vector, pKB△C_His_ as control). The bottom panel shows the amount of rRNA in each sample, stained with ethidium bromide as the loading control.

**Figure 3 viruses-11-00509-f003:**
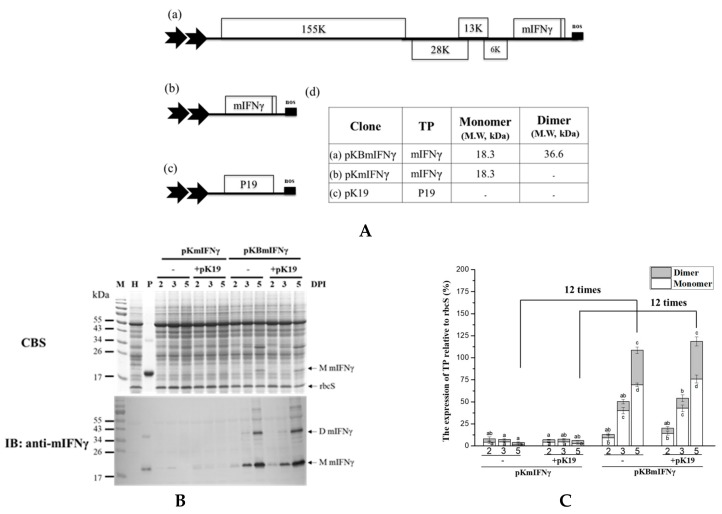
mIFNγ protein was efficiently produced by BaMV expression system. (**A**). *N. benthamiana* leaves were infiltrated with *A. tumefaciens* harboring pKBmIFNγ (a) and pKmIFNγ (b) (Nonviral vector) or co-infiltrated with pK19 (c) gene-silencing suppressor. The expected molecular weight of the respective TP indicated in the adjacent table (d) (**B**). Analysis of TP expression in inoculated leaves. Total protein extracts were prepared from infiltrated leaf tissue at 3 and 5 DPI, and analyzed by SDS-PAGE, followed by staining with CBS and IB analysis with anti-mIFNγ as primary antibody and goat-anti-rabbit IgG alkaline phosphatase conjugate as secondary antibody. M, Marker; H, Healthy leaf; P, Positive control, purified mIFNγ protein derived from *E. coli* (250 ng for CBS and 25 ng for IB). (**C**). Quantification with densitometer. Densitometric quantification showing the percentage values of monomer (M) or dimer (D) TP relative to RuBisCO Small subunit, rbcS. Statistical analysis was performed using ANOVA. *p*-value of <0.001 was considered significant.

**Figure 4 viruses-11-00509-f004:**
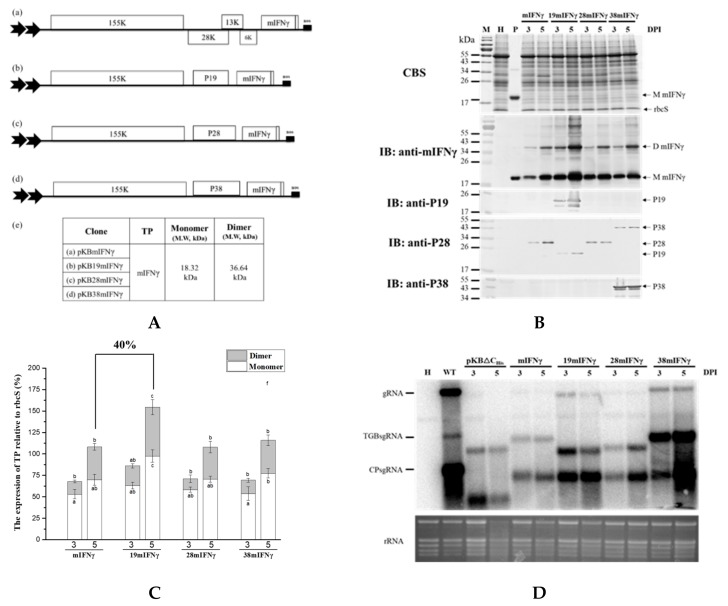
The effect of co-expressing different viral silencing suppressors on the accumulation of mIFNγ protein. (**A**). Schematic representation of BaMV-based systems for the co-expression of viral silencing suppressors and TP. In order to increase TP accumulation, mIFNγ was co-expressed with silencing suppressor protein P19, P28, and P38 individually, by replacing the TGBp coding region with that of the viral suppressors. These constructs were named pKBmIFNγ (a), pKB19mIFNγ (b), pKB28mIFNγ (c), and pKB38mIFNγ (d), with the expected molecular weight of the respective TP indicated in the adjacent table (e). (**B**). Analysis of TP expression in inoculated leaves. Total protein extracts were prepared from infiltrated leaf tissue at 3 and 5 DPI and analyzed as described above, except that the specific antibodies against mIFNγ, P19, TGBp1 of BaMV, or P38 were used as the primary antibodies in IB, and goat-anti-rabbit IgG alkaline phosphatase conjugate as the secondary antibody. M, Marker; H, Healthy leaf; P, Positive, purified mIFNγ protein derived from *E. coli* (250 ng for CBS and 25 ng for IB). (**C**). Quantification with densitometer. Densitometric quantification showing the percentage values of M or D TP relative to RuBisCO Small subunit, rbcS. Statistical analysis was performed using ANOVA. *p*-value of <0.001 was considered significant. (**D**). Northern blot analysis of wild-type or chimeric BaMV RNA in infiltrated leaves at 3 and 5 DPI. BaMV genomic RNA, and the subgenomic RNAs for triple gene block proteins (TGPsgRNA) and CP (CPsgRNA) were detected with a BaMV-specific probe, respectively (vector, pKB△C_His_ as control). The bottom panel shows the amount of rRNA in each sample, stained with ethidium bromide (EtBr) as the loading control.

**Figure 5 viruses-11-00509-f005:**
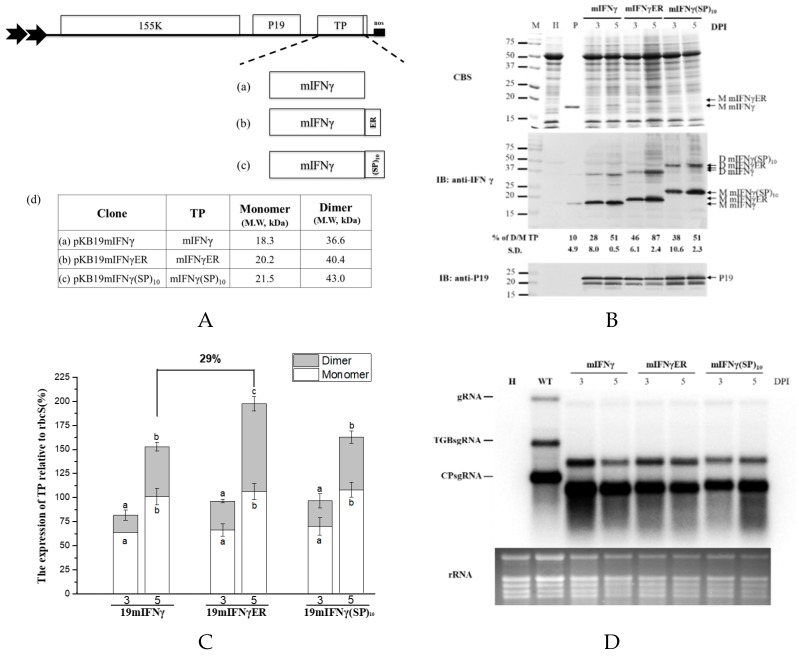
The stability and accumulation of mIFNγ was affected by different signals fusion. (**A**). Schematic representation of BaMV-based systems expressing TP. In order to increase TP stability, P19mIFNγ was fused with ER or (SP)_10_ signal. These constructs were named pKB19mIFNγ (a), pKB19mIFNγER (b) and pKB19mIFNγ(SP)_10_ (c), with the expected molecular weight of the respective TP indicated in the adjacent table (d). (**B**). Analysis of TP expression in inoculated leaves. Total protein extracts were prepared from infiltrated leaf tissue at 3 and 5 DPI and analyzed as described above, except that the specific antibodies against mIFNγ and P19 were used as the primary antibodies in IB, and goat-anti-rabbit IgG alkaline phosphatase conjugate as the secondary antibody. M, marker; H, healthy leaf; P, positive, purified mIFNγ protein derived from *E. coli* (250 ng for CBS and 25 ng for IB). (**C**). Quantification with densitometer. Densitometric quantification showing the percentage values of M or D TP relative to RuBisCO Small subunit, rbcS. Statistical analysis was performed using ANOVA. *p*-value of <0.001 was considered significant. (**D**). Northern blot analysis of wild-type (WT) or chimeric BaMV RNA in infiltrated leaves at 3 and 5 DPI. BaMV genomic RNA, and the subgenomic RNAs for triple gene block proteins (TGPsgRNA) and CP (CPsgRNA) were detected with a BaMV-specific probe, respectively (vector, pKB△C_His_ as control). The bottom panel shows the amount of rRNA in each sample, stained with ethidium bromide (EtBr) as the loading control.

**Table 1 viruses-11-00509-t001:** Expression levels of pKB19mIFNγ, pKB19mIFNγER, and pKB19mIFNγ(SP)_10_ in agroinfiltrated *N. benthamiana* leaves as determined by ELISA quantification.

Clone	DPI ^1^	%TSP ^2^	Yield (μg/g Fresh Weight)
19mIFNγ	3	1.5	59 ^a^ ± 1.6
	5	1.9	92 ^c^ ± 2.1
19mIFNγER	3	1.7	75 ^b^ ± 2.9
	5	2.5	119 ^d^ ± 0.8
19mIFNγ(SP)_10_	3	1.7	73 ^b^ ± 4.4
	5	2.3	111 ^d^ ± 5.8

^1^ DPI, days post-inoculation; ^2^ TSP, total soluble protein. Mean values with dissimilar superscripts (a,b,c,d) are significantly different at *p* < 0.001 level.

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
