# Peer review of "Production of Human IFNγ Protein in Nicotiana benthamiana Plant through an Enhanced Expression System Based on Bamboo mosaic Virus"

_viruses, 2019, doi:10.3390/v11060509_

Reviewer 1 Report

You manuscript describing the development of a virus base vector to produce human interferon gamma in plant as pharmaceutical is well described. However, adding 3 to 9 introducing line before each result paragraph made the results some quit more difficult to appreciate. All these small introductions need to move to the general introduction so that results are centralized on your work. The text to move is highlighted in the attached file.

Author Response

Manuscript ID: viruses-512068
Type of manuscript: Article
Title: Production of human IFNγ protein in Nicotiana benthamiana plant 
through an enhanced expression system based on Bamboo mosaic virus

Dear Reviewer 1,

Your manuscript describing the development of a virus based vector to produce human interferon gamma in plant as pharmaceutical is well described. However, adding 3 to 9 introducing line before each result paragraph made the results some quits more difficult to appreciate. All these small introductions need to move to the general introduction so that results are centralized on your work. The text to move is highlighted in the attached file.

Response: We are very thankful to the comments and suggestions of Reviewer 1. Accordingly, the highlighted short introductions in the Results section have been moved to either the Introduction section or Materials and Methods section in the revision, as shown by the “Track Changes” function of Word.

Thank you very much for the critical review and helpful suggestions. We have revised our manuscript accordingly.

Reviewer 2 Report

In the manuscript entitled “Production of human IFNγ protein in Nicotiana benthamiana plant through an enhanced expression system based on Bamboo mosaic virus” by Min-Chao Jing et al, authors featured some interesting contributions to plant molecular farming, by exploring possibilities of a plant viral vector based on Bamboo mosaic virus. The study raised interesting questions concerning the putative large-scale production of an important therapeutic protein, human INFγ. Corresponding assays are well conceived, viral-vector system was stepwise improved using different strategies and finally rendered not negligible expression levels of the target protein. This paper contributes improving the knowledge about the potential of large-scale production of target peptides in a cost-effective manner using plant cell systems and plant viral vector benefits. However, in the interest of this paper, authors are encouraged to check the glycosylation pattern of hINFγ produced in plants compared with that occurring in animals. It is also important to estimate how feasible could be scale up of a non-stable system, based on a CP deficient viral vector, and determine the heterologous gene stability expressed from this vector or in a particular context.

Specific comments:

1.        In addition to the work of Larsen et al. (2012) cited in Discussion, it would worthwhile that authors make some mention to other works in which high-yield expression of therapeutic proteins, as high as 30% total soluble protein, are obtained in Nicotiana benthamiana plants by using PVX-based vector [Mardanova et al., 2015].

2.        Authors should clearly point out what advantages and disadvantages have the use of expression based on BaMV-vector compares to similar systems. For instance, additionally to some papers cited in Discussion, compared to biologically active His-tagged INFγ already reported in plants, which accumulation reaches up to 6 % of total soluble protein (see Leelavathi and Reddy, 2003).

3.        Glycosylation pattern of recombinant IFNγ in plants should be deeper addressed, once expression systems can affect the IFNγ glycosylation and this post translational modification influences the half-life and activity of the molecule. This is particularly important here because available data about glycosylation IFNγ in plants are scarce (see Razaghi et al., 2016).

4.        Result 3.2: Why viral vectors of similar pathogens function in distinct manner? Please, provide some explanation, if any, to justify different levels of INF expression when BaMV-, PVX- or FoMV-based vectors are used. In this sense, it could be interesting check the stability of heterologous INFγ gene in the context of these three potexvirus-based vector.

5.        What differences in optimization codon applied to INFγ-1 and INFγ-2 constructs could explain higher accumulation level of the protein in this last one (in Figure 2B, pKBIFNγ-2 vs pKBIFNγ/pKBIFNγ-1)?

6.        In immunoblot assays: Could be possible that primary antibodies against mIFNγ recognize better mIFNγ than IFNγ, stablishing a false higher accumulation for the first one? Primary antibody should render signals with same intensity when same amounts of mIFNγ than IFNγ are loaded. Is this checked?

Minor comments:

1.        In general, annotations in CBS/blots, and especially in bar charts, are too much small. I suggest get better legibility increasing font sizes.

2.         In References Section, journal names must be given using corresponding abbreviations or its full names, but not one or the other, indistinctly. For instance, “Curr. Issues Mol. Biol”, “PLos Pathog” in some cases and “Plant biotechnology journal” or “Annual Review of Analytical Chemistry” in others.

3.        Line 85, Line 94 and Line 101: “Table 1” instead “Table S1”.

4.        Line 94: Dot missing before “The amplified...”

5.        Line 127: Providing the current Table 1, showing used primer list, as a Supplemental Table makes more sense.

6.        Line 144: close the parenthesis.

7.        Line 145: “Method S2” instead “supplementary methodS2”.

8.        Line 192: “Method S1” instead “method s1”.

9.        Line 197: Corresponding Figure must be referenced

10.    Line 214-227: In Figure 1 caption, it should be clarified what “a,b,c,d” meaning in bar chart.

11.    Line 247: “pKBINFγ-1” instead of “pKBINFγC-1”.

12.    Line 260: “(Fig. 2B and C)” instead “(Fig. 1B and C)”.

13.    Line 262: Extra dot at the end of the line.

14.    Line 291-292: Names for vectors (a) and (b) are swapped.

15.    In Fig. 4B: Apparently, there is a mistake naming the primary antibody used to detect the three RNA silencing suppressors.

16.    In Fig. 2C, 3C and 4C: Axis Y of bar chart, “expression of TP...” instead “expreesion of TP...”

17.    Lines 322-324: Sentence must be reformulated.

18.    Line 343: “[33]” instead “Bagheri et al., 2010”.

19.    Line 381-383: Sentence must be reformulated.

20.    Lines 357-358: Sentence must be reformulated.

21.    Line 405: Dot missing at the end of the line.

Author Response

Manuscript ID: viruses-512068
Type of manuscript: Article
Title: Production of human IFNγ protein in Nicotiana benthamiana plant 
through an enhanced expression system based on Bamboo mosaic virus

Dear Reviewer,

Thank you very much for the critical review and helpful suggestions. We have revised our manuscript accordingly. The revisions and our responses to the reviewers’ comments and questions are detailed as file:
